# Predictive Modeling and Computer Vision-Based Decision Support to Optimize Resource Use in Vertical Farms

KC Shasteen and Murat Kacira *
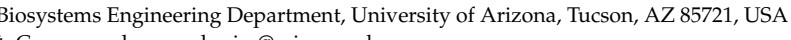

Biosystems Engineering Department, University of Arizona, Tucson, AZ 85721, USA
* Correspondence: mkacira@arizona.edu

**Abstract:** This study evaluated several decision-support tools that can be used to create a control system capable of taking advantage of fluctuations in the price of resources and improving the energy use efficiency of growing crops in vertical farms. A mechanistic model was updated and calibrated for use in vertical farm environments. This model was also validated under changing environmental conditions with acceptable agreement with empirical observations for the scenarios considered in this study. It was also demonstrated that lettuce plants use carbon dioxide ($CO_2$) more efficiently later in their development, producing around 22% more biomass during high $CO_2$ conditions during the fourth-week post-transplant than in the first week. A feedback mechanism using top-projected canopy area (TPCA) was evaluated for its ability to correlate with and provide remote biomass estimations. It was shown that for a given set of constant environmental conditions, a scaling factor of $0.21$ g cm$^{-2}$ allowed the TPCA to serve as a rough proxy for biomass in the period prior to canopy closure. The TPCA also was able to show deviation from expected growth under changing $CO_2$ concentrations, justifying its use as a feedback metric.

**Keywords:** computer vision; top-projected canopy area; predictive model; lettuce; dynamic control system; vertical farm; controlled environment agriculture; digital twin

## 1. Introduction

Vertical farming is the practice of growing on vertically stacked layers in a controlled environment using hydroponic techniques and artificial lighting [1,2]. Its purpose is to improve the efficiency of land use and water use while generating less waste, increasing the amount of control over the crop-growing environment, extending the growing season, reducing transport costs, and decreasing the risk associated with crop production. Despite its many efficiency benefits, vertical farming requires intense energy costs and has high capital and labor costs. Because of this, a great deal of effort has been invested in increasing the efficiency of resource use in such farms. Optimization of farm efficiency through both plant environmental setpoint research and facility design level decisions is critical to the financial viability of the industry, especially with regard to plant growth [3], HVAC design [4–6], and lighting technologies [7]. Much of the recent work to improve resource use efficiency, in agriculture more generally, focuses on the development of smart agricultural practices using modeling and decision-making tools such as artificial intelligence (AI) and distributed, Internet of Things (IoT) sensors and actuator networks for environmental control [8–10]. There are still critical challenges remaining unresolved with the integration of AI in greenhouse applications, including cost, gaps between research and development, technology commercialization, energy use, and the tradeoff between accuracy and computational speeds [8]. The anticipated economic benefits encourage the application of the IoT for optimized greenhouse environments and resource management, and any potential risks are unparalleled to the long-term benefits of commercial agriculture [9]. Physical automation of the growing environment has also been a focus of vertical farm development with substantial investment from agriculture technology sector companies. Automation

offers the promise of reducing high labor costs [11], especially around time-consuming activities, such as harvesting and transplanting.

The typical way of controlling plant growth in a vertical farm (VF) is to keep environmental conditions as constant as possible. However, this approach ignores the costs associated with maintaining the environment. Some essential resources such as electricity have times of the day, week, or year where their costs can be significantly lower. Other resources such as carbon dioxide ($CO_2$) can be used more efficiently at certain times in the development of a crop. Using this knowledge, along with models that predict plant growth under different environmental conditions, it might be possible to dynamically alter the environment around a crop to enhance resource use efficiency and lower operating costs. This approach is the basis for, and fundamental benefit of, dynamic control of the growing environment; however, an accurate plant growth model and feedback metrics are critical to success.

There are various possible ways to achieve cost savings using dynamic control. For instance, increasing either $CO_2$ or light can increase yields. However, because $CO_2$ is typically less expensive than lighting, exchanging $CO_2$ for light can produce significant cost savings. Likewise, heating ventilation and air conditioning (HVAC) economizers allow a building to cool and dehumidify outside air but tend to dilute any $CO_2$-enhanced air inside. A dynamic control system would allow the costs and benefits of these types of resource exchanges to be analyzed in real time as a crop grows. Many electrical providers have payment plans that discourage energy use during particular times; scheduling the lighting to activate only during cheaper times (off-peak hours) can save on electrical costs. Moreover, some utility plans provide discounts for allowing the electrical provider to throttle a facility's energy usage using real-time energy management (RTEM). Using RTEM, the provider would signal times and amounts by which electrical load should be reduced and the grower's control system would automatically dim the lights. Normally, variable lighting intensity would mean unpredictable growth for a crop; however, with dynamic control, any shortfalls in lighting could be offset with more light at times of lower demand such as nights and weekends. Similarly, in many places, the thermal gradient at night tends to produce favorable HVAC thermodynamics, meaning that not only is energy cheaper but the HVAC equipment is more efficient and placed under less load. Thus, the thermal environment of a VF can also be managed dynamically provided the plant's response to alterations in temperatures is accounted for in the predictive model.

Creating a constantly adapting environment inside a VF to minimize costs appears to be a novel idea. However, greenhouses (GHs) have always had variable environments. Recent research describes some of the methods used to deal with environmental variability, plant growth prediction, and profit optimization in GHs. In Hemming et al. [10], several artificial intelligence (AI) algorithms were tested using a combination of data from human workers and temperature, relative humidity (RH), light, and $CO_2$ sensors located in several GHs in order to develop strategies for controlling the GH environment. The controlled factors included ventilation, humidity, supplemental lighting, shading/blackout screen activation, heating, $CO_2$ concentration, irrigation, nutrient amendments, and pruning/harvesting rules: all were varied with the goal of optimizing profit, including price adjustment for fruit quality. Some AI algorithms had access to additional sensors, such as cameras, computer vision-based metrics, sap-flow and stem diameter sensors, and load cell lysimeters. Various strategies were offered by the AI algorithms, and all were able to outperform a team of human growers. The employed machine learning algorithms were based on conditional, rule-based algorithms, data-enabled predictive control (DeePC), long short-term memory networks (LSTM), bidirectional LSTM, reinforcement learning (RL), and imitation learning. Extra data from historical information, predictive crop modeling, and 24 h weather predictions were also included to inform some of the control strategies. Most AI design teams broke the decision-making process into strategic, tactical, and operational levels. Importantly, all teams had a human-in-the-loop for analysis and/or decision making, meaning that full direct AI control was not realized by any team. Rather, the AI

acted as an effective tool to guide strategic and tactical decisions. The study indicated that more data are needed to produce accurate growth models to aid AI in prediction and strategizing; that computer vision, especially, needs more development as a rich source of data; and that robotics development is key to realizing fully autonomous growing systems. Computer vision and deep learning algorithms have also been used for automated plant spacing decisions with lettuce crops toward the autonomous control of greenhouses [12]. However, it is also underlined that there is still a gap between research and commercial production conditions, and larger datasets are still needed to bridge the gap.

Feedback systems meant for GHs have been developed for measuring the temperature of plants remotely using thermography [13–15]. In these systems, temperature data, along with input solar radiation data, can be used to calculate the stomatal conductance of the canopy. This allows the system to infer information about plant water status, which can be used both to avoid irrigation deficiencies and to optimize the growing environment by maximizing the heating uniformity and heating efficiency for both the growing area and the facility as a whole. In a GH, this is performed by varying factors such as the degree of vent opening, shade cloth deployment, air temperature, absolute humidity, and $CO_2$ concentration. If the technique were adapted for a VF setting, where control would be enforced by an HVAC system, the knowledge of stomatal conductance could potentially prove useful for predicting conditions that might be conducive to tipburn, a common pathology of lettuce grown in VF.

There has also been substantial work carried out on using multispectral camera imaging techniques to characterize the water status of both field and GH crops [14,15]. These techniques take relative intensity values from various parts of the light spectrums (such as blue, green, red, far-red, and infrared) and apply mathematical formulae to create "spectral indexes". These are typically unitless values from 0 to 1 and indicate the water status of growing plant tissue. Popular indexes include the simple ratio (SR), the normalized difference vegetation index (NDVI), and the enhanced vegetation index (EVI), though there are many more that have been studied [16] and which might be adapted for use in VF. Unlike field-based measurements, where the amount of background soil in the image vs. leaf tissue has a strong influence on the signal [17], in the VF setting, the background can be easily removed from the image. Thus, in VF, the technique would rely primarily on measuring color changes due to slight shifts in pigment concentration within the plant, which occur as the water status changes. Phenotype features with basil crop height, weight, and leaf area with 3D reconstruction, leaf segmentation, geometric surface modeling, and deep network for estimation of crop weight have also been demonstrated in a VF setting [18].

For certain species, some of the indexes, especially together with other data, have been able to detect water stress [16]. Computer vision techniques using image texture and top-projected canopy area (TPCA) have also been shown to monitor crop growth and health status [19]. However, in the GH and field environments, these techniques are vulnerable to the high spatial and temporal variability in lighting conditions, and special treatment of the data is required to overcome the challenge. The presence of irregular background images and shadows can further complicate the application of the method [15]. The suitability of the technique is much improved in the well-controlled environment of a VF; however, the highly controlled nature of the VF also makes the incidence of stress much lower. Therefore, such a feedback system has limited application in VF using static setpoints but may be useful for dynamic control systems where continuous feedback is needed to verify how a crop is responding to changing environmental conditions. A dynamic control system requires three critical components. First is a predictive growth model, which allows the control system to know where a crop should be in its development for a given sequence of environmental conditions. Second is a plant-based feedback mechanism that allows the system to correct any deviation from the model predictions, allowing the system to know where the crop actually is. Third is a machine learning algorithm, which, together with the predictive growth model and cost functions for resources, gives the system a means of

evaluating future scenarios and finding an optimal sequence for minimizing costs while maintaining a target yield and harvest time. This study focuses on the first two of these critical elements.

Therefore, the objectives of this study were to (1) generate data needed to modify, validate, and test a predictive growth model for lettuce grown in a VF system, (2) implement and evaluate a computer vision system for its potential as a proxy for biomass and as a feedback control metric for a dynamic control system, and (3) to explore ways the predictive model can be used as a decision-support tool for co-optimizing environmental variables by exploring various what-if scenarios in which environmental conditions such as air temperature, light intensity, daily light integral (DLI), and $CO_2$ concentration were varied to help minimize input costs while maintaining acceptable yields.

## 2. Materials and Methods

During the experiments, a total of 10 crops were grown. Each crop consisted of 72 plants in the production system, and the central 40 plants, in the most uniform environment of the growing system, were used for data collection. This ensured that non-uniform (typically lower) light and airflow conditions in the outer regions did not bias the biomass accumulation of the plants used for data collection. Each crop was grown under different sets of both constant and variable environmental conditions, and direct measurements were made of root and shoot biomass at several points during crop development. Subsections of some crops were imaged with a multispectral camera throughout their development (although, ultimately, only data from the visible channels were used), and environmental conditions for all crops were monitored and recorded throughout.

### 2.1. Vertical Farm Facility and Instrumentation

The experiments were conducted in one of the growth chambers located at the vertical farm facility of the Controlled Environment Agriculture Center at the University of Arizona (the UAg Farm). The dimensions of the chamber floor were 3.68 m by 5.94 m and the chamber height averaged 3.8 m. The chamber contained two multilevel growing racks, each with three deep water culture growing levels where the lettuce was grown on floating foam rafts. Each level's growing area measured 2.44 m by 1.22 m, contained about 300 L of continuously recirculating nutrient solution and had two horizontal airflow fans that maintained air circulation in a race track pattern. Each rack had independent 1270 L nutrient reservoirs and shared a set of peristaltic acid/nutrient pumps (800-101-3014-17, ANKO, Bradenton, FL, USA). A single $CO_2$ sensor (GMP222, Vaisala, Vantaa, Finland) and transmitter (GMT220, Vaisala, Vantaa, Finland) measured $CO_2$ concentrations for the entire chamber. Light was provided by eight LED lighting fixtures (F3 Eclipse, Illumitex, Austin, TX, USA) per level, and was independently controlled for each level by a network-connected Infrastack lighting controller. The spectrum of these lights was 8% Blue (400–500 nm), 6% Green (500–600 nm), and 86% Red (600–700 nm), as measured by a spectroradiometer (PS-300, Apogee Instruments, Logan, UT, USA) just prior to the experiments. The temperature and RH of the aerial environment were monitored on one level of each rack with air temperature and RH probes (HMP60, Vaisala, Vantaa, Finland), which were enclosed in an aspirated and shielded sensor housing. Each rack also had its own set of aerial and root zone environmental sensors, whose signals were used to control the peristaltic pumps. The nutrient solution pH (HI 1001, Hanna, Woonsocket, RI, USA), EC (HI-EC 3001, Hanna, RI, USA; CDE-100-1 PT100, Omega, Egham, Surrey, UK), and dissolved oxygen (DO1200, Sensorex, Garden Grove, CA, USA) levels were also measured. The RH was controlled using a dehumidifier (CFT4.0D, Colzer, Yangzhou, China), which had its own internal sensors. In one level of one rack, a multispectral camera (RedEdge, MicaSense, Seattle, WA, USA) was fixed above the growing area, which collected image data as the crops developed. Except for the camera data, all sensor data were recorded using a datalogger (CR6, Campbell Scientific, Logan, UT, USA). Each rack also had a quantum sensor measuring photosynthetic photon flux density (SQ-120, Apogee Instruments, Logan,

UT, USA), which was calibrated to the spectrum of the growing lights and was used to map and verify instantaneous lighting intensity at the start of each experiment. During the experiments, these sensors were placed on one of the levels in each rack, and their data were used to confirm that the lighting schedule was followed. There was no detectable drop in lighting intensity between experiments.

### 2.2. Model Calibration Data Collection

In order to generate calibration data for a lettuce crop growth model, six crops of *Lactuca Sativa* L. (three Butterheads: cv. 'Pascal', cv. 'Seurat', and cv. 'Rex', and one Oakleaf: cv. 'Rouxaï') were grown under different environmental conditions. Because the model was designed to accumulate plant biomass under varying environmental conditions, it was important to validate the model's ability to estimate biomass under time-varying conditions. Therefore, one crop was grown under constant setpoints, while the five others were grown under conditions that changed midway through their development. Specifically, for these five crops grown simultaneously in the same chamber, the atmospheric $CO_2$ concentration was changed 12 days after transplant. All other factors were held constant.

The data collected from these six crops were used to calibrate and modify a model previously used for GH environments. The model parameters were adjusted to minimize the error of the model-predicted biomass against the actual-measured biomass across the entire crop development period. This allowed the model to make reasonably accurate predictions of biomass at any time in a crop's development. Table 1 summarizes the cultivars and environmental conditions for these crops. These environmental conditions were selected based on data previously gathered in the VF about when tipburn would appear, to avoid the condition and keep the onset of tipburn from influencing the model [20].

**Table 1.** Target environmental setpoints for carbon dioxide concentration [$CO_2$], photosynthetic photon flux density (PPFD), daily light integral (DLI), air temperature (Temp), and vapor pressure deficit (VPD) for crops 1–6 used to generate calibration data for the predictive growth model. DAT stands for days after transplanting.

| Crop | Condition | Cultivar | [$CO_2$] 0–12 DAT (ppm) | [$CO_2$] 12–28 DAT (ppm) | Photo Period (h) | PPFD ($\mu$mol $m^{-2}$ $s^{-1}$) | DLI (mol $m^{-2}$ $day^{-1}$) | Photo Period Temp. (°C) | Dark Period Temp. (°C) | VPD (kPa) |
|---|---|---|---|---|---|---|---|---|---|---|
| 1 | static | 50:50 mix, Pascal (green), and Seurat (red) | 400 | 400 | 16 | 201 | 11.6 | 23 | 19 | 1.0–1.2 |
| 2 | varying | Pascal | 400 | 900 | 16 | 201 | 11.6 | 23 | 19 | 1.0–1.2 |
| 3 | varying | Rex | 400 | 900 | 16 | 217 | 12.5 | 23 | 19 | 1.0–1.2 |
| 4 | varying | Rex | 400 | 900 | 16 | 250 | 14.4 | 23 | 19 | 1.0–1.2 |
| 5 | varying | Rouxaï | 400 | 900 | 16 | 217 | 12.5 | 23 | 19 | 1.0–1.2 |
| 6 | varying | Rouxaï | 400 | 900 | 16 | 250 | 14.4 | 23 | 19 | 1.0–1.2 |

Pure $CO_2$ was injected into the facility from externally stored high-pressure steel cylinders (capacity 7.08 $m^3$). The nutrient solution used for all crops was a modified Haogland's solution specifically for lettuce crops. For mature crops, a full-strength solution was used and had a target electrical conductivity (EC) of 1.8 dS $m^{-1}$. For propagation, a half-strength solution was used. In both cases, the target potential of hydrogen (pH) was 5.8–6.0, and pH control was performed using nitric acid.

Seedlings for these crops were germinated and matured using a half-strength nutrient solution in a GH for roughly 14 days before being transplanted into the growing boards in the vertical farm facility. The production continued for 28 days after transplanting until the final harvest. Sample plants were taken and measured from the crops at regular intervals.

For crops 3 through 6, these samples were carried out in seven-day intervals; however, for crops 1 and 2, samplings were clustered closer together (4 days apart) to accommodate comparisons of biomass to top-projected canopy area (TPCA), as measured by the computer vision system. All crops were sampled, measured, and completely harvested at 28 days after transplant (DAT). The sample measurements included fresh mass (FM) and dry-mass (DM) measurements of both shoot and root biomass. The sample size was eight (n = 8) randomly selected plants for all but crop 1, where the sample size was ten (n = 10) randomly selected plants (5 'Pascal' and 5 'Seurat'). Measurements of FM were taken immediately after harvest, and plant tissues were then dried for at least 3 days (or until all water had been driven off and the mass had stabilized) at 65 °C in order to measure DM. Table 2 shows the sampling times and sizes for each crop.

**Table 2.** Crop sampling schedule for model calibration data.

| | Time of Sampling in Days after Transplanting (DAT) | | | | | | | | | | |
|---|---|---|---|---|---|---|---|---|---|---|---|
| | 0 | 7 | 10 | 12 | 14 | 16 | 18 | 20 | 21 | 22 | 28 |
| **Crop** | Sample Size (Number of Plants Sampled, n) | | | | | | | | | | |
| **1** | | | 10 | | 10 | | 10 | | | 10 | 10 |
| **2** | 8 | | | 8 | | 8 | | 8 | | | 8 |
| **3–6** | 8 | 8 | | | 8 | | | | 8 | | 8 |

Due to a very similar growth habit between the cultivars, a Student's *t*-test was used to determine the statistical significance of differences between the masses of 'Pascal' and 'Seurat' plants in crop 1. These differences were not significant at the 95% confidence level, so these plant data were aggregated into a single sampling (n = 10) thereafter.

### 2.3. Top-Projected Canopy Area (TPCA) Experiment

A second experiment was performed during the development of crops 1 and 2. The purpose of the experiment was to obtain top-projected canopy area (TPCA) data so that TPCA could be evaluated as a proxy for biomass and its suitability as a feedback signal for an automated control system could be assessed. Because the atmospheric environment was varied for this experiment, these two crops were grown consecutively rather than concurrently. The salient variable in this experiment was $CO_2$ concentration. Crop 1 was grown for 28 DAT with a constant $CO_2$ of 400 ppm, while crop 2 was grown at 400 ppm for the first 12 days and then at 900 ppm until final harvest at 28 DAT. Images were taken of a subsection of the canopy during the development of both crops. A single multispectral camera (RedEdge, MicaSense, Seattle, WA, USA) was used to gather images in five different spectral channels (blue, green, red, far-red, and near-infrared) every 15 min. After image capture, custom-built image processing software (coded in C++ and with OpenGL) first segmented the leaf area from the background image and then counted the number of pixels of leaf contained in a representative portion of each image. A reference area, whose dimensions were known, was used to find the proportion of area to pixels at the depth of the growing plane. This proportion was then multiplied by the number of leaf pixels counted earlier to calculate the area of the visible leaf, as viewed from above (the TPCA) in each image for both crops 1 and 2. At the same time, biomass measurements were taken, as described in the previous section. Together, these data sets were graphed for comparison and analysis.

### 2.4. Model Validation Experiment

In a third, subsequent experiment, the model's ability to make predictions under changing environmental conditions was evaluated. This experiment also tested the effects of timing on $CO_2$ treatment. Four additional crops (crops 7–10) of the cultivar 'Rouxaï' were grown as $CO_2$ conditions were varied by week. For crops 7 through 10, the $CO_2$ was

kept high for the 4th through 1st week, respectively, and $CO_2$ remained at near-ambient conditions for all other weeks. The growing period for these crops was staggered across 7 weeks, such that all crops shared the same $CO_2$ treatment but at different times in their development (see Figure 1). The lower $CO_2$ setpoint was 450 ppm and the high setpoint was 900 ppm.

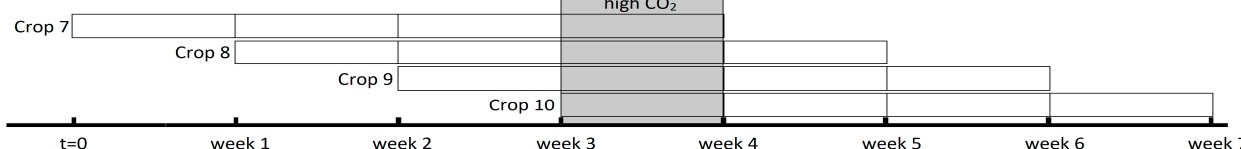

**Figure 1.** Experimental design for the validation experiment. Room $CO_2$ was kept constant at 450 ppm until the beginning of week 3 when it was changed to 900 ppm. At the beginning of week 4, it was dropped again to 450 ppm. In this way, each crop experienced the treatment at a different time in its development. Samples were taken for all levels at the beginning and end of each week of development.

Biomass measurements of shoot FM and DM were taken at transplant and 7-day intervals from all crops until harvest. Additionally, at 0 DAT, a measurement of root FM and DM was taken of crop 7 to gather more complete data for the root allocation function of the predictive model of crop biomass. The sample size for all of these measurements was eight (n = 8).

Since having a consistent starting point was critical for this experiment, propagation was performed in the VF facility rather than in a GH. Seedlings were placed in an environment with a PPFD of 100 µmol m$^{-2}$ s$^{-1}$ for 16 h a day for 3 days to germinate; the PPFD was then adjusted to 200 µmol m$^{-2}$ s$^{-1}$. Seedlings were given half-strength modified Hoagland's nutrient solution for the duration of this period, and on the day of the transplant, their fresh and dry shoot mass was measured and used as a starting point for the model.

The aerial environmental setpoints for this experiment were much the same as for the previous two, except for $CO_2$ concentration and timing. Table 3 lists the aerial environmental set points for crops 7–10.

**Table 3.** Target aerial and lighting environmental setpoints for crops 7–10. Listed are the growing conditions for the aerial and lighting environment for the crops used to test and validate the predictive growth model.

| Crop | Condition | Cultivar | Low [CO$_2$] Setpt. (ppm) | High [CO$_2$] Setpt. (ppm) | CO$_2$ High during (WAT) | Photo Period (h) | PPFD (µmol m$^{-2}$ s$^{-1}$) | DLI (mol m$^{-2}$ day$^{-1}$) | Photo Period Temp. (°C) | Dark Period Temp. (°C) | VPD (kPa) |
|---|---|---|---|---|---|---|---|---|---|---|---|
| 7 | varying | Rouxaï | 450 | 900 | 4th | 16 | 201 | 11.6 | 23 | 19 | 1.0–1.2 |
| 8 | varying | Rouxaï | 450 | 900 | 3rd | 16 | 201 | 11.6 | 23 | 19 | 1.0–1.2 |
| 9 | varying | Rouxaï | 450 | 900 | 2nd | 16 | 201 | 11.6 | 23 | 19 | 1.0–1.2 |
| 10 | varying | Rouxaï | 450 | 900 | 1st | 16 | 201 | 11.6 | 23 | 19 | 1.0–1.2 |

Abbreviations: [CO$_2$]—carbon dioxide concentration; WAT—weeks after transplant; setpt.—setpoint; PPFD—photosynthetic photon flux density; DLI—daily light integral; Temp.—temperature; VPD—vapor pressure deficit; the precision of the PPFD measurements was to the tenths place.

## 3. Results and Discussion

In this section, the crop biomass predictive model is introduced along with an explanation of modifications made to it. The results of model simulations based on the measured environmental conditions during the experiments are then presented, and model prediction accuracy from the validation experiment is discussed. Furthermore, area measurements for

the TPCA are presented, and its utility as a feedback tool and limitations of edge effects are discussed. Future directions of research are considered for both the predictive model and computer vision-based biomass measurements.

### 3.1. Model Calibration

The model used in the current research was adapted from a physiological model, using differential equations that were validated by van Henten in 1994 [21]. Starting from an initial mass at transplant, the model uses input environmental conditions (specifically, air temperature, $CO_2$, and light intensity) to iteratively predict changes in structural (cell walls/proteins) and non-structural (sugars/starches) biomass on a dry-mass basis. Physiological parameters for the model such as carboxylation conductance, leaf area ratio, light use efficiency, and others were gathered by van Henten from the literature and were, in most cases, general estimates for lettuce as a species. Thus, differences between cultivars and differences across development were not reflected in the model results. Figure 2 contains a simplified flow chart showing model dynamics.

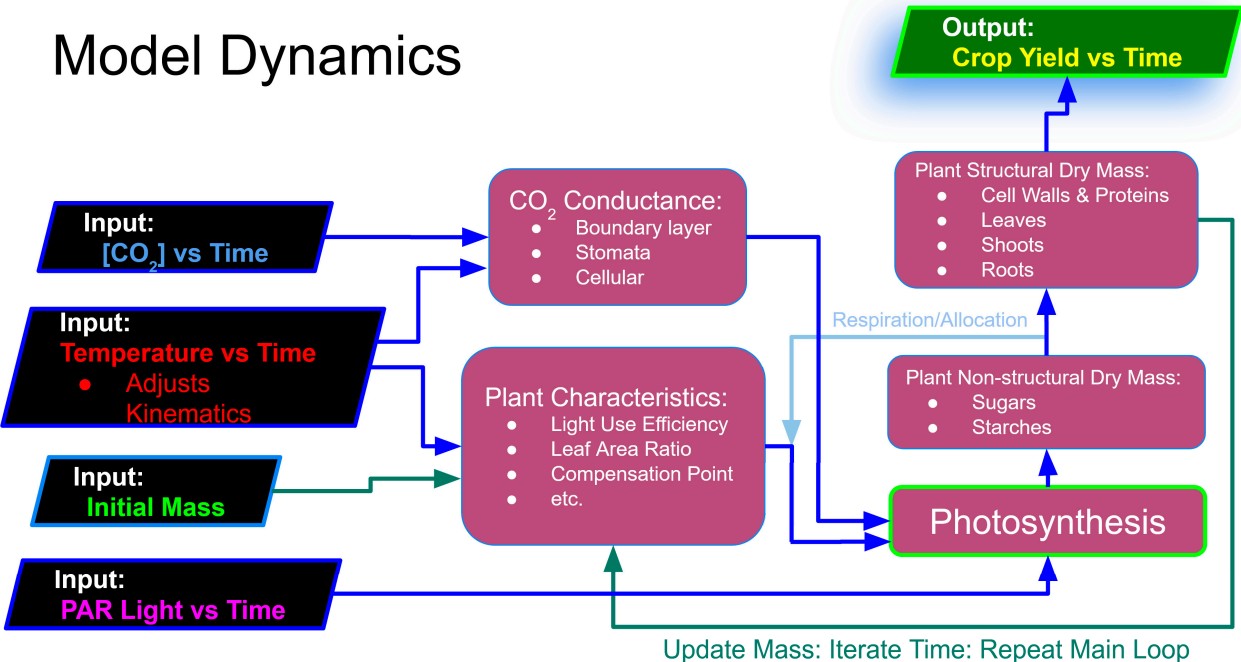

**Figure 2.** Model dynamics for the predictive growth model.

A custom Python code was written, implementing the model described above, to predict crop yield outcomes. Five adjustments were made to van Henten's originally presented model to increase its accuracy compared to the observed biomass based on observations from crops 1–6. First, an updated value for $CO_2$ density, $c_\omega$, was used; this change increased both the precision and accuracy of the value. The new value for the density of $CO_2$ was $1.893 \times 10^3$ g m$^{-3}$, and is given for a temperature of 15 °C and a pressure of $1.01325 \times 10^5$ Pa (interpolated from Table 5 of Anwar and Carroll, 2016) [22]. Second, a "yield factor", $c_\beta$, representing growth losses due to respiration and the synthesis of nonstructural material, was changed from a value of 0.8 to 0.72. This alternate value is also suggested as a possibility by van Henten based on a measurement by van Keulen et al. (1982) [23], and the change significantly lowers biomass predictions throughout development, improving accuracy across the entire time domain. Third, the $CO_2$ compensation point, $c_\Gamma$, was increased until a good match between model predictions and observed data was found. This increased its value from 40.0 ppm to 71.5 ppm. Current research has shown that the $CO_2$ compensation point is a function of light intensity at the time of leaf development [24]. Because the model's original parameters were for

a GH setting, where sunlight is the light source, higher intensities were probable, and the lower value of 40.0 ppm was likely to be accurate. However, in the VF setting, the lighting intensities tend to be lower, thus a higher value for compensation point is more reasonable. He et al. [24] showed compensation points of around 60 ppm for lettuce grown at 240 μmol m$^{-2}$ s$^{-1}$; therefore, the model's slightly higher value of 71.5 ppm was considered reasonable. This change had a larger effect on predictions when input environmental $CO_2$ was low, and was particularly useful for increasing accuracy for crop 1 where $CO_2$ was low throughout. Fourth, a function for root allocation was added, replacing a constant $c_\tau$. Root allocation is the ratio of root DM to total plant DM. This was performed to improve the accuracy of the model across time, as it was overestimating shoot mass in early development and underestimating it in later development. The original constant value of 0.15 was based on measurements of soil plants, while the new function was based on data taken from crops 1–7 (grown in DWC) and has values between 0.0788 and 0.203, peaking in week one and then tapering off exponentially toward harvest. The new function for root allocation makes use of the gamma distribution from the field of Statistics (see Figure 3).

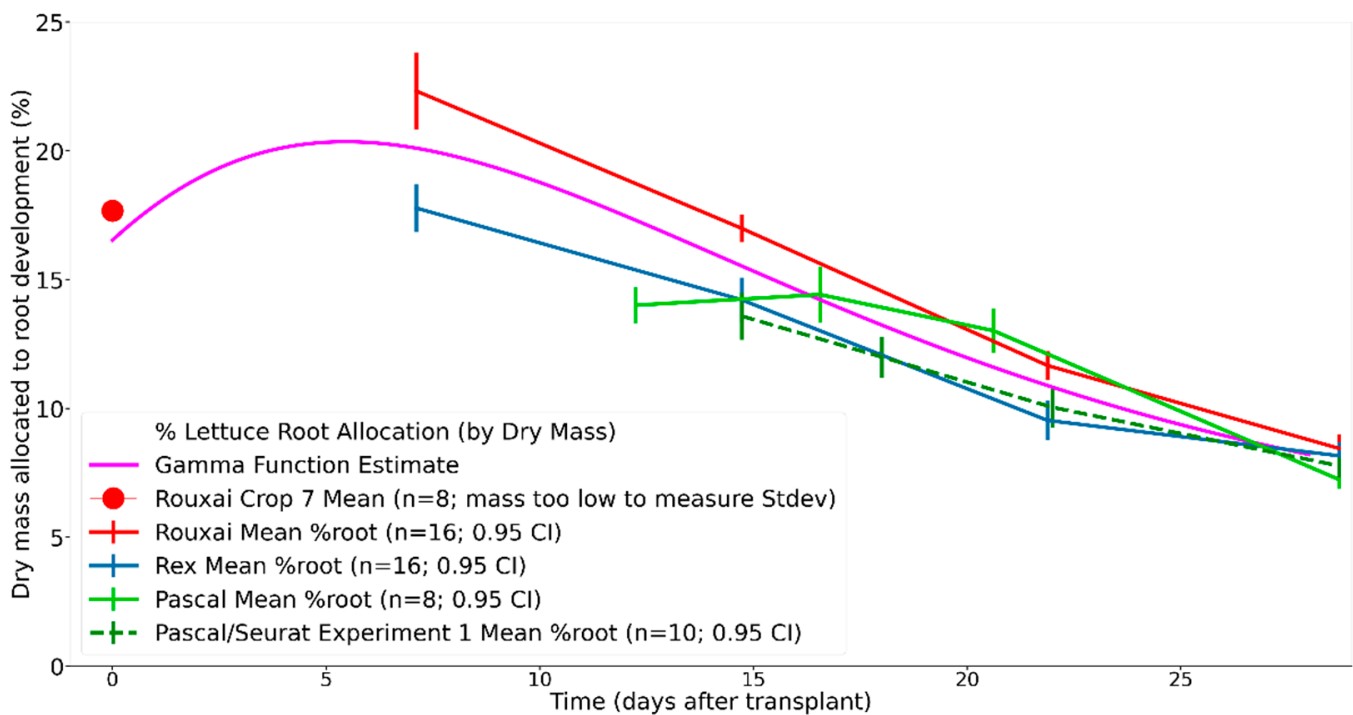

**Figure 3.** Root allocation function added to the predictive model, replacing a constant. This function estimates the proportion of DM allocated to root production. The measured values, which guide the shape of the functions, are also plotted.

The fifth and final modification to the model was the addition of a function for estimating water content; it is called the dry-mass ratio function. This addition allowed the model to convert its output from DM to FM. The function was fitted to the observed data from crops 1–7. Similarly, to the root allocation function, the dry-mass ratio function makes use of the gamma distribution (Figure 4).

Additionally, of note, a conversion factor for "sun and sky in daylight" of 4.57 μmol m$^{-2}$ s$^{-1}$ per W m$^{-2}$ [25] was used to convert measurements of photosynthetic photon flux density (PPFD) to the native power density of the model equations.

Measured and predicted fresh lettuce head biomass is shown in Figure 5, along with the $CO_2$ concentration, light intensity, and air temperature used to model the biomass predictions. Predictions are based on a measurement of initial biomass at DAT 0. The $CO_2$ and air temperature measurements were directly measured from the aerial environ-

ment in which each crop was grown, while the light intensity was derived from actual measurements of the on/off state of the lights, but the magnitude of the signal has been adjusted to reflect the measurements before each experiment began. A 10% reduction in light magnitude was simulated and is shown to account for reduced diffuse lighting as the canopy developed. It was found that the predictions of the modified model were in good agreement with measured fresh shoot biomass.

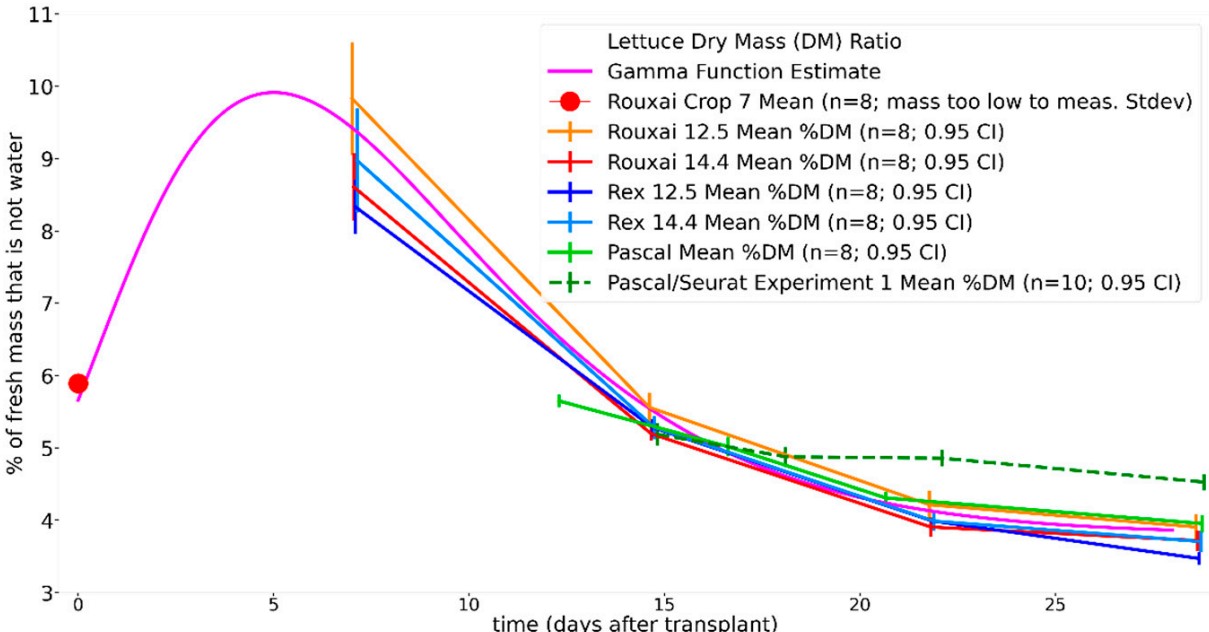

**Figure 4.** Dry-mass ratio function (aka the "not-water" function). This function was added to the predictive model for estimating the amount of FM on a moment-by-moment basis. The measured values, which guide the shape of the function, are also plotted.

For most applications, evaluating the accuracy of the model is less simple than merely determining the degree or rate of error of the final prediction against the final harvest mass. Because the model is iterative, for the model to be effective in time-varying circumstances, it must be accurate across many points along the course of a crop's development. This makes it difficult to represent the accuracy of the model with a single figure. Moreover, since the new functions are a generalized estimate of data from several cultivars, it will necessarily not include the physiological variation from specific cultivars. In these regards, compromises are inevitable; for instance, while the final mass estimates for 'Rex', 'Seurat', and 'Pascal' are quite good, the final mass estimates for Rouxaï fail to reflect an observed paradoxical mass decrease at higher light levels (possibly related to tipburn). Likewise, finding reasonable physiological parameters, which allowed good final estimates, also required all modeled predictions of early mass to be at the low end of the 95% confidence interval. Thus, the predictive accuracy can be characterized as being decent in generalizing the mid- and final masses of lettuce, while the early-mass predictions trend low, though still within the interval of 95% confidence.

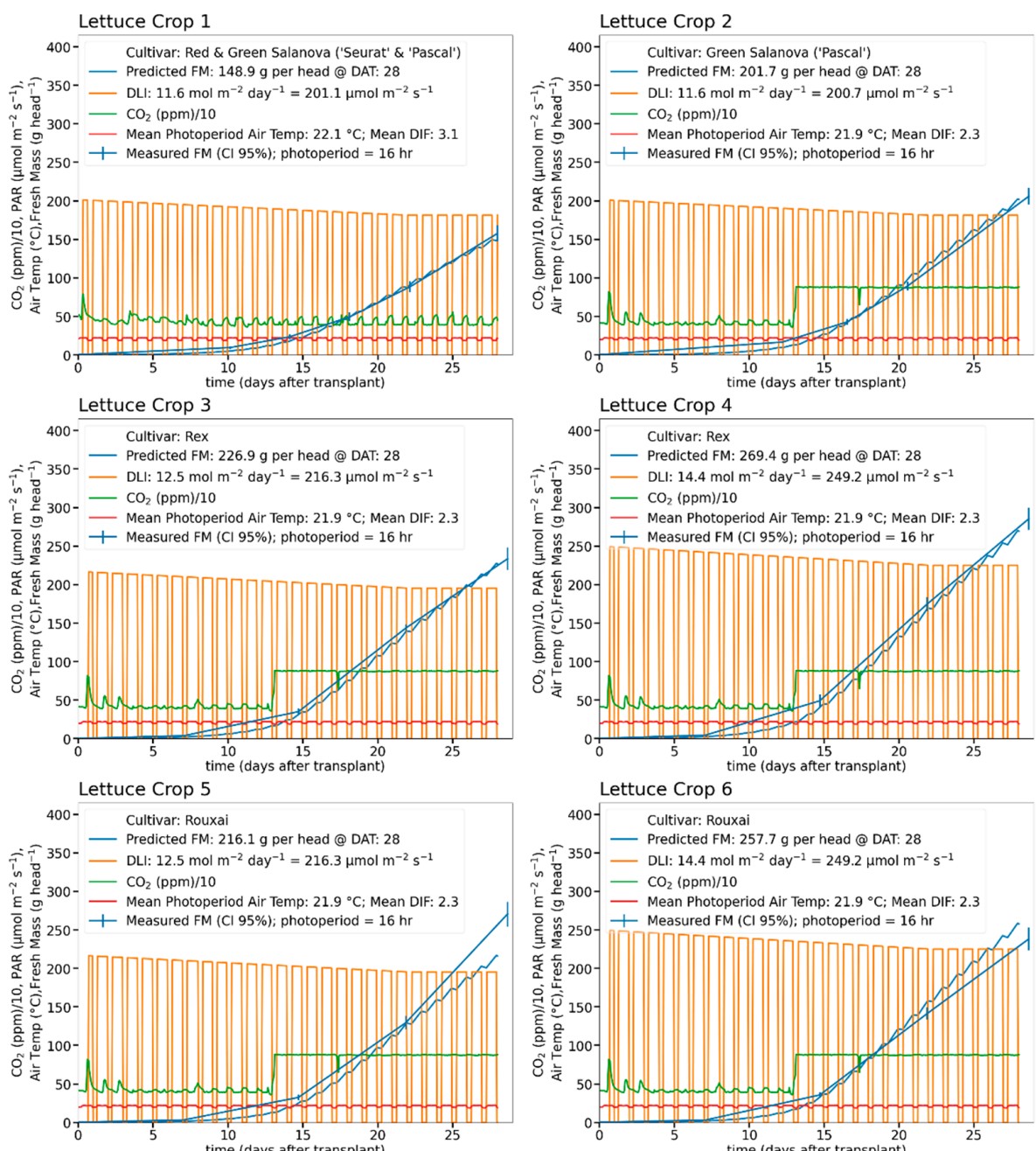

**Figure 5.** Observed and predicted fresh biomass for crops 1–6, along with the $CO_2$ concentration, light intensity, and air temperature used to model the predictions. These environmental values reflect the actual conditions in which the crops were grown.

### 3.2. Top-Projected Canopy Area

Computer vision calculated the top-projected canopy area (TPCA) and measured the FM for two crops in different sets of environmental conditions, one with $CO_2$ kept constant at 400 ppm and the other with $CO_2$ increased from 400 ppm to 900 ppm at day 12 after transplant (Figure 6).

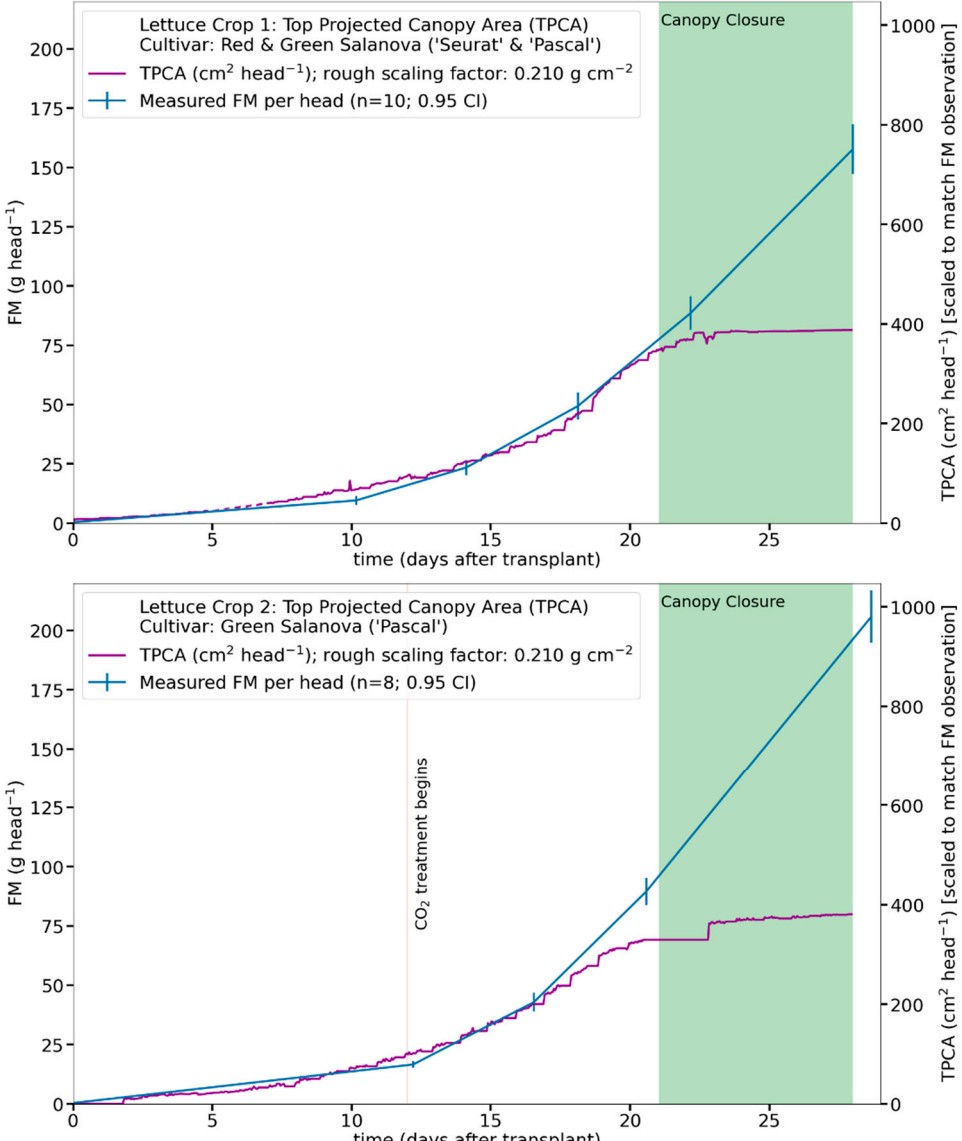

**Figure 6.** Computer vision calculated the top-projected canopy area (TPCA) and measured fresh mass (FM): (**top**) Crop 1 where $CO_2$ was kept constant; (**bottom**) Crop 2, where $CO_2$ was increased from 400 ppm to 900 ppm at day 12 after transplant.

As can be seen from the data, a very close estimation of biomass can be achieved during the first 18–20 DAT of development by multiplying the TPCA graph by a scalar value. In this study, the value of 0.21 g cm$^{-2}$ was determined to provide good results for both crops grown under the same lighting conditions. However, it is important to note that because the rate of canopy expansion is dependent on factors such as light intensity, light spectrum, and cultivar, this scaling factor should be adjusted depending on the conditions to be used for crop production. Thus, instead of being scalar, the conversion from area to mass should be a function of these other factors, especially in systems where these factors can be varied.

Additionally, TPCA was evaluated for its utility as a feedback tool. Figure 7 shows the two TPCA graphs together along with the time of $CO_2$ treatment. Before the treatment, the rates of canopy expansion indicated by TPCA are nearly equal, and they begin to diverge almost immediately upon the start of treatment. This divergence continues to expand until about 18 DAT. After 20 DAT, the closure of the canopy causes the two signals to converge as a horizontal expansion of each plant once again (and the TPCA measurement itself) begins to be influenced by its neighbors. This indicates that, prior to canopy closure, the

TPCA has the potential to be used for detecting deviations in growth due to changes in $CO_2$ conditions. It also shows that the TPCA may be a useful part of a feedback system for a dynamic control system.

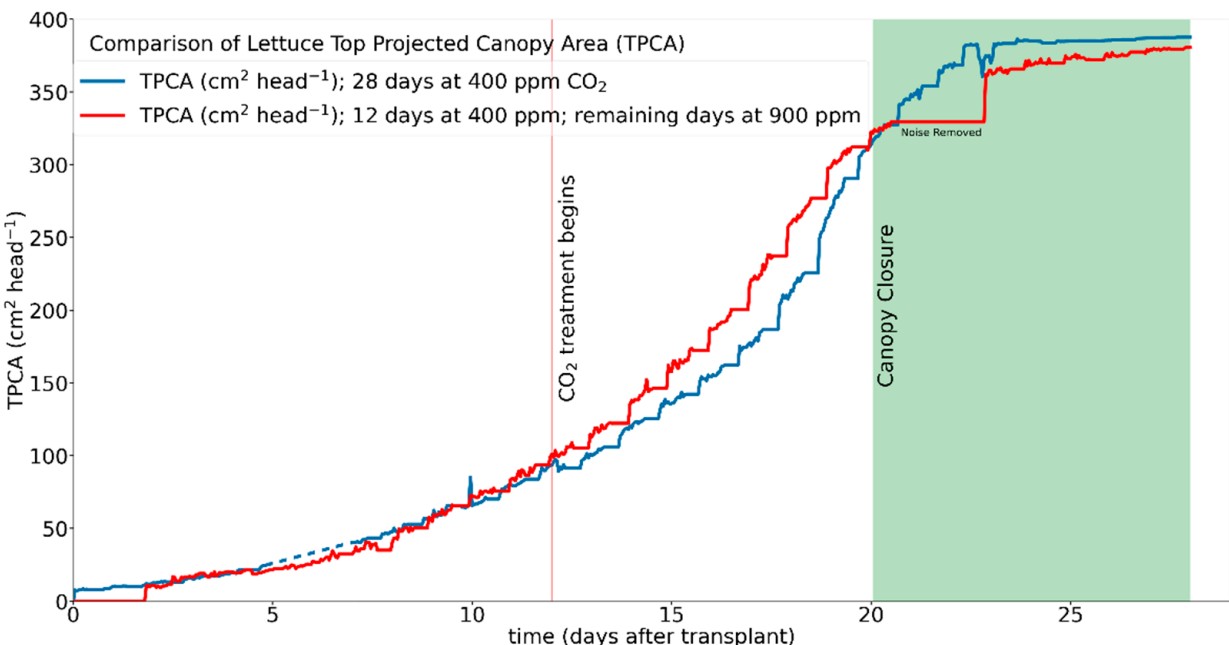

**Figure 7.** TPCA comparison graphs for two crops: one grown at a constant 400 ppm $CO_2$ concentration and the other where $CO_2$ was increased from 400 ppm to 900 ppm at day 12. The TPCA appears to diverge about the same time as the treatment begins.

In order to reduce edge effects on the TPCA signal, the sampled crop image area was reduced, and only a single planting hole's worth of image area was considered rather than the 4–6 plants originally envisioned. Because of this, the standard deviation of the signal was not characterized, and it remains uncertain just how much deviation in the TPCA signal is required to demonstrate off-nominal conditions. If the 95% confidence interval for biomass measurements is scaled and then superimposed on the TPCA signal (as in Figure 6), it appears that divergence may require several additional days to confirm for the n = 8 sample size. This could be improved, however, by imaging a larger area.

*3.3. Model Validation*

In this experiment, $CO_2$ was kept high (900 ppm) during only a single week of a crop's development, and low (450 ppm) at all other times. The model's predictions for crop performance are shown in Figure 8. The predictions indicated that more developed crops were able to make better use of the increased $CO_2$ levels and accumulated more biomass.

These predictions were supported by measurements from crops 7 through 10. The measured results for the final harvest of these crops are shown in Figure 9, along with the model's predictions. Predictions for fresh biomass fell within the confidence intervals for observed values for all crops except for the crop where $CO_2$ treatment was during the first week after transplant. This is possibly because the model slightly underpredicts growth in the very early crop development stage. The poor accuracy for early predictions during high $CO_2$ in week 1 of the model then possibly caused later predictions to overshoot.

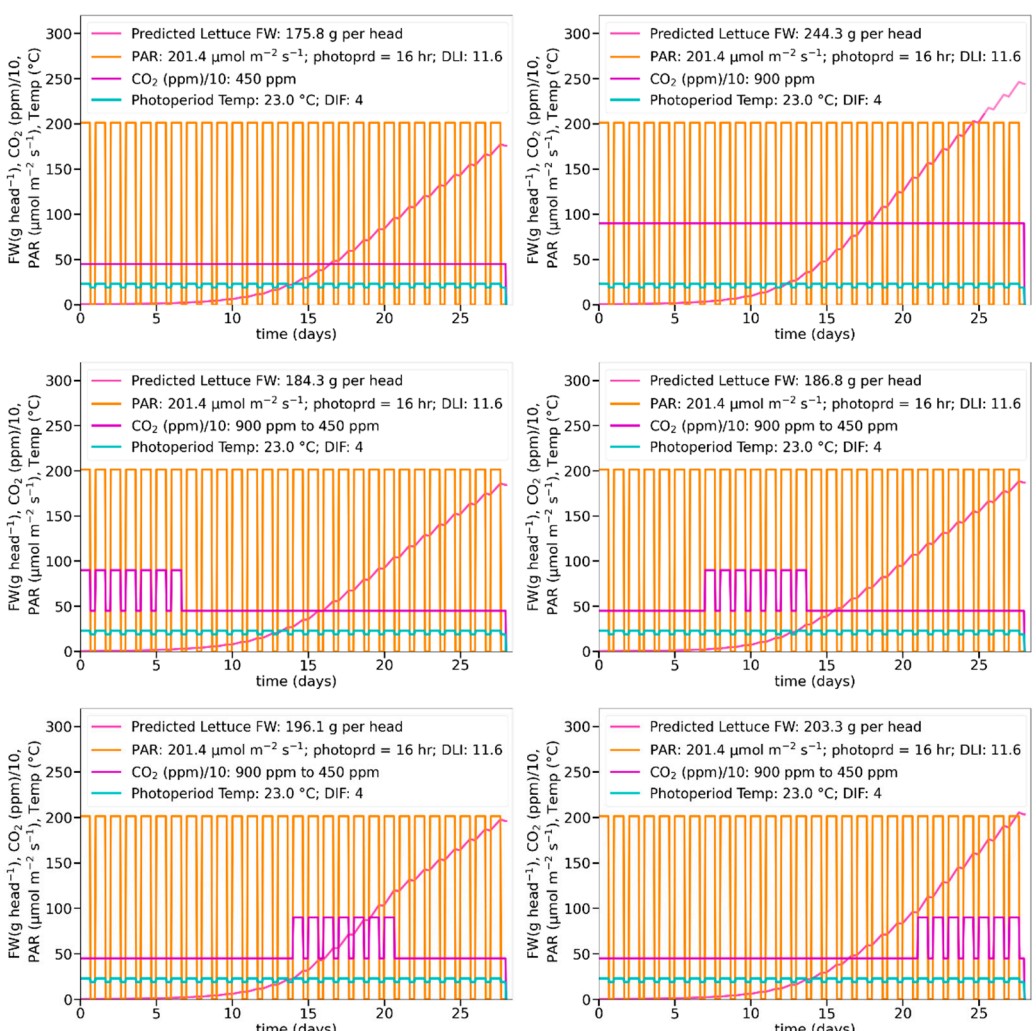

**Figure 8.** Predicted growth functions under various $CO_2$ scenarios. Shown also are the environmental conditions and the final predicted head FM for each scenario: (**top left**) $CO_2$ is held at 450 ppm for all 4 weeks; (**top right**) $CO_2$ is held high at 900 ppm for all 4 weeks; (**middle left**) $CO_2$ is kept high only in the 1st week after transplant; (**middle right**) $CO_2$ is kept high only in the 2nd week after transplant; (**bottom left**) $CO_2$ is kept high only in the 3rd week after transplant; (**bottom right**) $CO_2$ is kept high only in the 4th week after transplant.

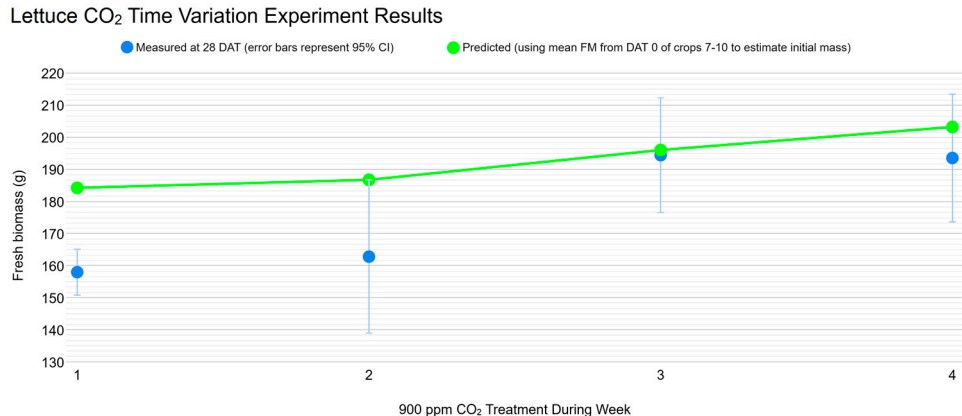

**Figure 9.** Measured (blue dots) and the predicted (green dots) FM at 28 DAT for the $CO_2$ time-varying validation experiment.

Overall, the model performed well and was able to predict the empirically observed upward trend of fresh biomass for high $CO_2$ in later development. Comparing the final observed biomass from the crop grown with high $CO_2$ in the fourth week after transplant to the crop in the first week after transplant, there was a 22% increase in head FM.

## 4. Discussions

### 4.1. Co-Optimization of Environmental Variables

The parameters influential to plant growth, such as light intensity, air temperature, carbon dioxide concentration, photoperiod, and DLI, can be controlled precisely. Co-optimization of these variables can contribute to significant resource savings in VF systems. As the validated model was able to predict growth trends under time-varying conditions, it was also useful for making predictions under constant environmental conditions. For example, Figure 10 provides 3D visualizations of how different sets of constant environmental conditions with air temperature, carbon dioxide, and DLI affect crop growth and yield. The updated model presented in this study and the information on the effects of environmental variables on crop yield and growth can be used as part of a decision-support system for identifying more efficient environmental setpoints for growing lettuce (even absent dynamic control), evaluating various what-if scenarios leading to significant resource savings in the development of advanced environmental controls, and also as a tool for improving experimental designs for research applications.

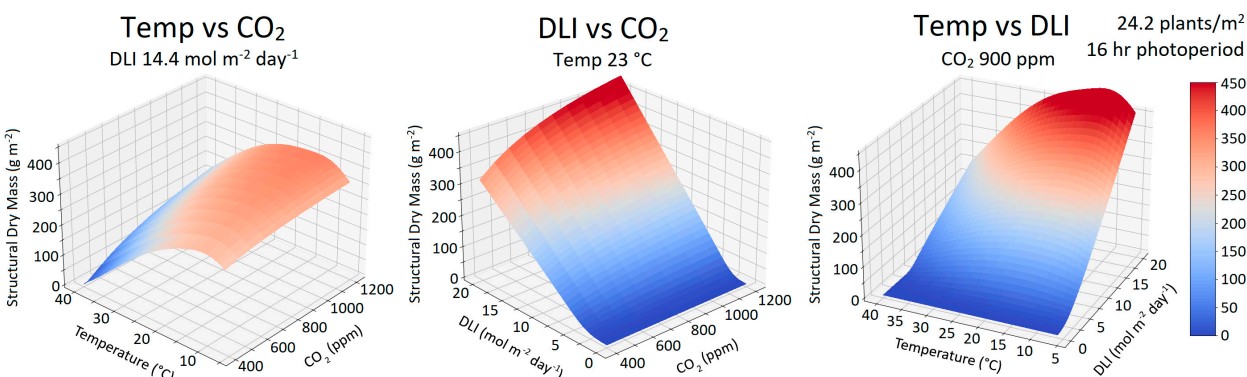

**Figure 10.** The model-predicted lettuce DM yield on day 28 after transplant with respect to $CO_2$ concentration, temperature, and daily light integral.

The model predictions showed that dry biomass yield was optimal at air temperatures in the vicinity of 12 °C and that tripling $CO_2$ from the ambient can drive this optimal temperature up to 15 °C. This is consistent with the current understanding of the physiology of photosynthesis, where increased $CO_2$ improves efficiency by reductions in photorespiration. In contrast, increases in DLI were predicted to lower the optimal air temperature, which shifted from an optimum near 20 °C at very low light levels to an optimum of about 12 °C at higher levels. Notably, these temperatures are lower than those often used for lettuce production. These increases in yield, though small, are likely caused by reduced metabolic sugar consumption outpacing reductions in photosynthesis at lower temperatures. It should also be noted that many of these optimums are quite far outside the validated range, and caution should be taken when drawing conclusions. Because the gradient near the validated range is relatively small, solutions for nearby values can still be expected to be somewhat accurate, but larger or longer departures can include a great deal of uncertainty. For evaluating the curves, the exact value of the optimum is less important than the trend surrounding it.

When holding the temperature constant, the relationship between yield and $CO_2$ is always positively correlated; however, there are diminishing returns as $CO_2$ levels increase. This is consistent with known relationships between growth and $CO_2$. Meanwhile, yield vs. DLI remains quite linear, as light intensity increases the fresh biomass yield increases

proportionally. It should be emphasized that while this trend would continue indefinitely in the modeled crops, real crop growth would find a limit at extreme $CO_2$ concentrations and DLI (e.g., from tipburn or $CO_2$ toxicity). Therefore, it is important to restrict model use to values near those that have been previously validated (i.e., the central regions of the graphs of Figure 10).

*4.2. TPCA as a Proxy for Biomass*

This study evaluated the rate of crop expansion using the TPCA as a proxy for biomass, particularly with respect to $CO_2$. However, because the rate of leaf expansion and overall compactness of lettuce plants is affected by both light intensity and spectrum, it is critical that the influence these two factors have on both biomass and the TPCA expansion is characterized before the TPCA can become an accurate proxy for biomass. This is especially true in dynamic lighting environments. Ideally, the effect of light on the TPCA should be mapped out over a range of different intensities and for various commonly used light spectrums. With this data, it would then be possible to create approximate mappings from the TPCA to biomass under variable conditions. This would allow the TPCA-to-biomass scaling factor used in this research to be replaced with a conversion factor that is a function of light intensity and spectrum and allows enhanced accuracy when correlating between TPCA and biomass.

Future research on TPCA should seek to measure a greater number of plants, ideally the entire canopy. Smaller sampling areas are subject to distorted signals from anomalous growth and can have significant edge effects from plants growing beyond (or into) the field of view. With a complete orthomosaic of the growing area (prior to canopy closure), each plant can then be segmented and measured independently. The TPCA then can be represented as a canopy-wide mean and standard deviation. This improves the TPCA's use as a diagnostic tool because unexpected changes in the TPCA's standard deviation can indicate non-uniform growth. This fact might also be used by a control system to indicate the presence of disease or equipment failure and allow the system to signal a need for human intervention.

The onset of canopy closure leads to a saturation of the TPCA signal. For this reason, some other metrics should be able to supplement the TPCA. Several other computer vision factors may be measured and could aid in the real-time, remote assessment of biomass, for example, plant height, image texture, and spectral indexes. It is likely that taking these factors together and including information about cultivar, light intensity, and light spectrum, experiments could be run to create biomass data that could be used to train a neural network to create very accurate estimations of biomass. Uncompressed, at 15 min intervals, the image data from five channels required about 33 gigabytes of storage per experiment. After downsampling to 1 h intervals, image processing to remove background, and video compression and encoding, this was reduced to about 24 megabytes, making long-term storage of growth videos of several thousand crops very affordable. Thus, a project to create a training database for machine learning using TPCA, image texture, or spectral data as a metric could be a viable and useful tool for improving growth modeling. Future research should explore this possibility due to its strong potential for use as not only a feedback tool for dynamic control but also a tool for growth modeling, stress detection, and disease management.

*4.3. Future Model Improvements*

There are still various constants and growth parameters within the modified crop biomass yield prediction model which were estimated using a single constant value, representing an average across all time. For better accuracy, these variables can be replaced with functions of time in a manner similar to the root allocation function that was added in the current study. Parameters such as leaf area ratio, respiratory and synthesis losses, and extinction coefficient can be considered for such modification as they are the most likely to change significantly with time. The existing time-based functions, including the ones

developed in the current study, should be improved in their temporal resolution particularly in the first week of crop development, since differential equations are quite sensitive to initial conditions. This can be performed by repeating the calibration experiments and taking biomass measurements while sampling daily during the first week of crop growth after transplanting.

Furthermore, other parameters should also be evaluated for how their values change under various environmental conditions. These parameters, rather than being functions of time, can become functions of the environmental inputs. For instance, leaf area ratio, extinction coefficient, respiration rate, and $CO_2$ compensation point are likely to change based on lighting intensity. The $CO_2$ compensation point especially has been shown to change by almost 50% as lighting setpoints decrease from outdoor intensities to levels found in VF [24]. Accurate characterization of these changes could greatly improve model accuracy.

Moreover, the range of environmental conditions used for the validated model in the current study can be expanded by growing additional calibration crops at higher light intensities, other air temperature ranges, and differing $CO_2$ levels. Exploring additional cultivars can be worthwhile to ensure the model is representative across the various genotypes. To improve model performance during early crop development, more frequent sampling of biomass in future calibration crops can be considered, for instance, sampling every 1 to 2 days in the first week and every 4 to 7 days thereafter. At higher growth rates, the tipburn problem can be characterized and the risk of tipburn, its severity, and the time of onset of symptoms can become an output of the crop growth model. This would allow the grower or the environmental control system to avoid yield loss due to tipburn.

The model could further benefit from the addition of equations for modeling the interaction of stomata, RH, and leaf boundary layer characteristics. In this study, the model calculated carboxylation conductance from two constants and an equation of temperature. Two of these constants, stomatal and boundary layer conductance, can be formulated as functions of the environmental conditions. For instance, a plant will adjust its stomatal conductance based on factors such as light intensity (particularly blue light), VPD, and $CO_2$; moreover, boundary layer conductance is a function of air-current speed. Modifying the model to accommodate these changes would not only improve accuracy but also provide a mechanism for predicting (and avoiding) conditions that might lead to tipburn within lettuce crops, thus an environmental control strategy can be established to mitigate the tipburn issues.

## 5. Conclusions

In conclusion, this study updated and validated a mechanistic model that is capable of predicting the yield of lettuce crops grown in a VF system. Several decision-support tools and environmental control strategies were evaluated and presented that can support creating a control system capable of taking advantage of fluctuations in the price of resources and improving the energy use efficiency of growing crops in VFs. The TPCA can serve as a rough proxy for biomass in the period prior to canopy closure and can be used as a feedback mechanism for environmental controls and lettuce crop growth. It was shown that lettuce plants use $CO_2$ more efficiently later in their development, producing around 22% more biomass with $CO_2$ enrichment applied during the fourth-week post-transplant than in the first week, thus offering the potential for dynamic $CO_2$ enrichment controls and cost savings. The evaluation of environmental conditions with air temperature, $CO_2$, and DLI on crop growth and yield using the validated model revealed various co-optimization strategies with these environmental variables that can lead to resource and cost savings for lettuce crop production, with potential further application opportunities for other crops in VF systems.

**Author Contributions:** Conceptualization, K.S. and M.K.; methodology, K.S. and M.K.; software, K.S.; validation, K.S. and M.K.; formal analysis, K.S. and M.K.; investigation, K.S.; resources, K.S. and M.K.; data curation, K.S. and M.K.; writing—original draft preparation, K.S.; writing—review and editing, K.S. and M.K.; visualization, K.S.; supervision, M.K.; project administration, M.K.; funding acquisition, M.K. All authors have read and agreed to the published version of the manuscript.

**Funding:** This research was sponsored in part by the USDA-SCRI project, award number: 2019-51181-30017.

**Institutional Review Board Statement:** Not applicable.

**Informed Consent Statement:** Not applicable.

**Data Availability Statement:** The datasets used and/or analyzed in this manuscript are available from the corresponding authors by email.

**Acknowledgments:** The core biomass growth modeling code was written by Ying Zhang, a graduate student from the Kacira Lab, based on the growth model outlined by van Henten (1994), and was extensively modified with crop parameters suited to the current study, and graphical output functions were produced by KC Shasteen. The experiments and the project were conducted at the University of Arizona's Vertical Farm facility located at the Controlled Environment Agriculture Center (UAg Farm). The authors acknowledge Tamara Friedman for her assistance in nursing, transplanting, harvesting, and collecting data on crops and in operating and maintaining the UAg Farm, and Neal Barto for his technical assistance.

**Conflicts of Interest:** The authors declare no conflict of interest. The funders had no role in the design of the study; in the collection, analyses, or interpretation of data; in the writing of the manuscript; or in the decision to publish the results.

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
