# Peer review of "Predictive Modeling and Computer Vision-Based Decision Support to Optimize Resource Use in Vertical Farms"

_sustainability, doi:10.3390/su15107812_

Round 1
Reviewer 1 Report
Dear Authors,
Please carefully check and solve all observations as formulated (below)
and sent to the editors of the journal:
<<Dear Editors,
As the authors stated, this paper (id sustainability-2168525, entitled “Predictive Modelling and Computer Vision Based Decision Support to Optimize Resource Use in Vertical Farms”) aims to evaluate and develop tools required to create a DCS (Dynamic Control System) by varying the environment in which lettuce is grown and by knowing the costs required to maintain that environment. Moreover, the authors claim that a predictive growth model was modified and validated for lettuce in vertical farms and this model was calibrated against experimental results from several cultivars, and model predictions were tested under time-varying conditions. Moreover, the authors state that by using this model, feedback-based control, and Machine Learning, it will be possible to co-optimize CO2, temperature, and light intensity to predict future environments that save electrical energy and growing expenses. Additionally, the model can evaluate set-point-based growth, aid experimental design, and help strategize on more optimal growing environments in vertical farms without DCS. Finally, the authors conclude that their research predicted and demonstrated that lettuce plants use CO2 more efficiently later in their development.
After reading this paper, I found some issues waiting to be dealt with.
I will start with the format ones. Then I will continue with those related to this paper’s substance/content if applicable.
I mention below the following:
-
English language and style issues - Grammarly (https://app.grammarly.com) on default settings (American English, Set Goals: Audience=Knowledgeable, Formality=Neutral, Domain=General) detected only for the text block resulting from the concatenation of Title+Abstract+Keywords +First paragraph of Conclusions and Recommendations+Acknowledgments:
8 correctness issues (critical alerts),
but 20 more complex ones (advanced suggestions).
Consequently, the resulting Grammarly overall/total score as reported by this online tool was 75 (left edge of Fair i.e. >=70, and still not Good i.e >=80, or Very Good/Excellent i.e. >=90) out of 100 (max) for this five-component sample above. Still, since the authors do not appear to be native English speakers, I suggest a comprehensive revision of the English language and style for the entire article using Grammarly or another specialized tool;
-
The paper must follow all the instructions of the journal precisely indicated at: https://www.mdpi.com/journal/sustainability/instructions;
-
The authors must avoid ending some sections/subsections with formulas, figures, tables, or other components (e.g., Fig. 9 just before Conclusions and Recommendation, and no explanatory text after). The authors are required to check the entire manuscript for similar issues;
-
The authors must additionally ensure that all figures have the required resolution (minimum 1000 pixels width/height, or a resolution of 300 dpi or higher). Most of the figures suffer in these terms;
-
The legend texts/titles of some of the figures and tables are too large (more than two lines). The authors should move part of these text blocks in the main text of the manuscript, just near the first reference to them. The authors are also required to check the entire manuscript for such issues;
-
There is a high number of figures (10) in this manuscript. Some of them which are considered by the authors not essential for understanding the main flow of ideas in the manuscript must be moved to the Appendix section. If this section is not existing, the authors must create one;
-
The List of Abbreviations at the end of the manuscript is needed;
-
The same for the Limitations corresponding to the research approach used;
-
The authors are also required to include more explanations and precise details about the standard view of the accuracy values (>=70% and <80%-fair models; >=80% and <90%-good models; >=90%-very good/excellent models) and their application here. They should provide more references to scientific papers where this topic is considered and the accuracy intervals are precisely defined;
-
The final list of just 13 references is far from enough. It indicates that many important related contributions in journal papers have not been cited in the Related Work/Literature Review section;
-
Moreover, the section dedicated to the interpretation of the results (Results and Discussions - clearly emphasized this way) needs more development and cited references to similar/different results (already published articles in highly rated scientific journals);
-
The authors must understand that replicability as a fundamental principle in science (https://doi.org/10.1007/s10516-021-09610-2 https://doi.org/10.1038/nature.2016.20504 ) starts with data and it is not a fad but a necessity. Therefore, they should insert in the Data Availability Statement section at the end of the manuscript all precise links to all data providers’ / own datasets used when testing the entire approach. In this context, I wonder, what really means the following paragraph “The datasets used and/or analyzed in this manuscript are available 887 from the corresponding authors on reasonable request by e-mail.” the authors included in the Data Availability Statement?;
-
Following this scientific principle above, the authors should provide full details about the software (including the precise name of the provider and version number of all the tools/apps) and also complete details about the hardware they used to test their approach and obtain the results presented in this manuscript;
-
The authors must also triangulate using many methods and techniques (https://doi.org/10.1038/d41586-018-01023-3) and perform many rounds of random cross-validations (https://doi.org/10.1007/978-0-387-39940-9_565, https://doi.org/10.1016/j.procs.2021.08.128) in order to prove their approach and the corresponding research results obtained are robust;
-
The Conclusions section should be restructured in a way that would better emphasize the authors’ contributions. Right now this section is more appropriate for being included in the Discussions / Result and Discussions one;
Thank you for the opportunity to read and check this manuscript!>>
Author Response
Responses to Reviewer Comments
Reviewer 1
Comment: As the authors stated, this paper (id sustainability-2168525, entitled “Predictive Modelling and Computer Vision Based Decision Support to Optimize Resource Use in Vertical Farms”) aims to evaluate and develop tools required to create a DCS (Dynamic Control System) by varying the environment in which lettuce is grown and by knowing the costs required to maintain that environment. Moreover, the authors claim that a predictive growth model was modified and validated for lettuce in vertical farms and this model was calibrated against experimental results from several cultivars, and model predictions were tested under time-varying conditions. Moreover, the authors state that by using this model, feedback-based control, and Machine Learning, it will be possible to co-optimize CO2, temperature, and light intensity to predict future environments that save electrical energy and growing expenses. Additionally, the model can evaluate set-point-based growth, aid experimental design, and help strategize on more optimal growing environments in vertical farms without DCS. Finally, the authors conclude that their research predicted and demonstrated that lettuce plants use CO2 more efficiently later in their development.
Response: The reviewer has a good understanding of the study objectives and made a good summary of key information and contributions made to the literature base don the outcomes of the study presented in the manuscript.
Comment: After reading this paper, I found some issues waiting to be dealt with. I will start with the format ones. Then I will continue with those related to this paper’s substance/content if applicable. I mention below the following:
Comment: English language and style issues - Grammarly (https://app.grammarly.com) on default settings (American English, Set Goals: Audience=Knowledgeable, Formality=Neutral, Domain=General) detected only for the text block resulting from the concatenation of Title+Abstract+Keywords +First paragraph of Conclusions and Recommendations+Acknowledgments: 8 correctness issues (critical alerts), but 20 more complex ones (advanced suggestions).
Consequently, the resulting Grammarly overall/total score as reported by this online tool was 75 (left edge of Fair i.e. >=70, and still not Good i.e >=80, or Very Good/Excellent i.e. >=90) out of 100 (max) for this five-component sample above. Still, since the authors do not appear to be native English speakers, I suggest a comprehensive revision of the English language and style for the entire article using Grammarly or another specialized tool.
Response: It is difficult to follow the approach that the reviewer discussed here to check the English of the paper. For instance the Title, Keywords, and Acknowledgments do not follow a grammatical structures with complete sentences, thus their including in the English and style check with grammar can lead to inaccurate analysis. Furthermore, it is very unfortunate that the reviewer made a wrong assumption indicating that “the authors of the paper do not appear to be native English speakers,” demonstrating an unconscious bias.
Comment: The paper must follow all the instructions of the journal precisely indicated at: https://www.mdpi.com/journal/sustainability/instructions;
Response: We reviewed the journal guidelines for authors and followed these guidelines within the revised paper.
Comment: The authors must avoid ending some sections/subsections with formulas, figures, tables, or other components (e.g., Fig. 9 just before Conclusions and Recommendation, and no explanatory text after). The authors are required to check the entire manuscript for similar issues;
Response: The paper was revised considering the comment made by the reveiwer.
Comment: The authors must additionally ensure that all figures have the required resolution (minimum 1000 pixels width/height, or a resolution of 300 dpi or higher). Most of the figures suffer in these terms;
Response: The resolution and image quality of the figures were improved to follow the journal guidelines.
Comment: The legend texts/titles of some of the figures and tables are too large (more than two lines). The authors should move part of these text blocks in the main text of the manuscript, just near the first reference to them. The authors are also required to check the entire manuscript for such issues;
Response: The paper was revised considering the comment made by the reveiwer. However, having reviewed other published articles in the Sustainability journal, we also notice that there are articles with 8-9 lines of text with information provided in the captions to further elaborate information in the tables/figures for the readers.
Comments: There is a high number of figures (10) in this manuscript. Some of them which are considered by the authors not essential for understanding the main flow of ideas in the manuscript must be moved to the Appendix section. If this section is not existing, the authors must create one;
Responses: The figures considered in the paper are required to clearly articulate the work performed in this study.
Comment: The List of Abbreviations at the end of the manuscript is needed;
Response: No abbreviation list was added as the Sustainability template does not call for a list of abbreviations. Furthermore, having reviewed the Sustainability articles. We did not find any containing a list of abbreviations.
Comment: The authors are also required to include more explanations and precise details about the standard view of the accuracy values (>=70% and <80%-fair models; >=80% and <90%-good models; >=90%-very good/excellent models) and their application here. They should provide more references to scientific papers where this topic is considered and the accuracy intervals are precisely defined. The final list of just 13 references is far from enough. It indicates that many important related contributions in journal papers have not been cited in the Related Work/Literature Review section;
Response: Additional references have been added and cited in the literature review and related work sections to enhance the discussions in the revised paper.
Comment: Moreover, the section dedicated to the interpretation of the results (Results and Discussions - clearly emphasized this way) needs more development and cited references to similar/different results (already published articles in highly rated scientific journals);
Response: Additional references have been added and cited to enhance the discussions in the revised paper.
Comment: The authors must understand that replicability as a fundamental principle in science (https://doi.org/10.1007/s10516-021-09610-2 https://doi.org/10.1038/nature.2016.20504 ) starts with data and it is not a fad but a necessity. Therefore, they should insert in the Data Availability Statement section at the end of the manuscript all precise links to all data providers’ / own datasets used when testing the entire approach. In this context, I wonder, what really means the following paragraph “The datasets used and/or analyzed in this manuscript are available from the corresponding authors on reasonable request by e-mail.” the authors included in the Data Availability Statement?;
Response: As authors, we clearly state our willingness to share the data sets used and analyzed in this study requested by others in the Data Availability section. The reviewer can refer to other published manuscripts in the Sustainability Journal, which also used same/similar statements in the Data Availability sections for access to the data sets used and analyzed.
Comment: Following this scientific principle above, the authors should provide full details about the software (including the precise name of the provider and version number of all the tools/apps) and also complete details about the hardware they used to test their approach and obtain the results presented in this manuscript.
Response: A detailed presentation and discussions of the hardware, software and code used in the study were presented in the materials and methods section in the revised paper.
Comment: The authors must also triangulate using many methods and techniques (https://doi.org/10.1038/d41586-018-01023-3) and perform many rounds of random cross-validations (https://doi.org/10.1007/978-0-387-39940-9_565, https://doi.org/10.1016/j.procs.2021.08.128) in order to prove their approach and the corresponding research results obtained are robust;
Responses: We made a detailed presentation of the materials and methodologies used in the study. This study is unique with experimental data generated from 4 lettuce crop varieties, with 6 crop cycles, under various environmental conditions and used for model validations, and research conducted in a large-scale crop production setting in a vertical farming system. And, to our knowledge, the study is one of a kind in the literature considering the scale of the data sets generated and used for the intended model updating and co-optimization of environmental variables that can be used to develop advanced and resource conserving environmental control strategies.
Comment: The Conclusions section should be restructured in a way that would better emphasize the authors’ contributions. Right now this section is more appropriate for being included in the Discussions / Result and Discussions one;
Response: The structuring of paper was significantly improved. A discussion and a new conclusions section were created presenting key outcomes and findings of the study. We discussed and presented several decision support tools and environmental control strategies that can be considered for practice and applications in vertical farming systems, which can support creating a control system capable of taking advantage of fluctuations in the price of resources and improving the energy use efficiency of growing crops in vertical farms.
Comment: Thank you for the opportunity to read and check this manuscript!
Response: We thank the reviewer for the recommendations provided which helped to strengthen our paper.
The following are addition revisions made in the revised paper:
- line 39, added indent
- line 210, table reformatted, reduced font size from 12 to 10, changed font to palatino linotype, adjusted column sizes, removed asterisk, shortened post-text, added vinculums to indicate precision
- line 240, reduced font size from 12 to 10, changed font to palatino linotype
- line 244, removed carriage return
- line 316, table reformatted, reduced font size form 12 to 10, changed font to palatino linotype, adjusted column sizes, removed asterisk, shortened post-text, added vinculums to indicate precision
- line 747, changed section number from 1 to 5
- line 857, added [12] to citation, removed parenthetical citation
- replaced all figures with higher resolution versions
- changed font to palatino linotype for all figure text
- removed whitespace and changed images to anchor to paragraph
- moved text from after figure 4 to between figures 3 and 4, added paragraph commenting on accuracy after figure 4
Respectively submitted,
Murat Kacira
Corresponding author
Reviewer 2 Report
Interesting topic and research contribution are also good. Title "Predictive Modelling and Computer Vision Based Decision Support to Optimize Resource Use in Vertical Farms" is effective enough to read the paper.
Few suggestions are given to improve the overall work:
(1) Introduction Section is well explained but it will help the readers if more clarification is given about author’s research contributions. List your research contribution in bullets for readers. Also, one paragraph can be added regarding the organization of this paper.
(2) Some typo error about Section number of the Conclusion Section.
(3) Section “Conclusions and Recommendations” is too length. This may be reorganized. Also, Table 1 & 2 may be given uniform font size.
(4) What is the significance of this work?
(5) Mention useful new knowledge about a completely novel idea or it is modifications in existing methods by applying a new algorithm.
(6) How you work is useful to provide directions for future research.
(7) Authors are suggested to review more new and relevant research to support their research contribution. The following references are recommended for possible consideration: Gadekallu, T. R., Srivastava, G., Liyanage, M., Iyapparaja, M., Chowdhary, C. L., Koppu, S., & Maddikunta, P. K. R. (2022). Hand gesture recognition based on a Harris hawks optimized convolution neural network. Computers and Electrical Engineering, 100, 107836
Author Response
Responses to Reviewer Comments
Reviewer 2
Comment: Interesting topic and research contribution are also good. Title "Predictive Modelling and Computer Vision Based Decision Support to Optimize Resource Use in Vertical Farms" is effective enough to read the paper. Few suggestions are given to improve the overall work:
Response: Thank you!
Comment (1): Introduction Section is well explained but it will help the readers if more clarification is given about author’s research contributions. List your research contribution in bullets for readers. Also, one paragraph can be added regarding the organization of this paper.
Response: We thank the reviewer for the constructive review and suggestions. The introduction section was improved with discussions on current state of the art on environmental controls and strategies applicable to vertical farming and the objectives and focus on the current study and paper were clarified on the introduction section as suggested.
Comment (2): Some typo error about Section number of the Conclusion Section.
Response: Section numbering error was corrected in the revised version.
Comment (3): Section “Conclusions and Recommendations” is too length. This may be reorganized. Also, Table 1 & 2 may be given uniform font size.
Response: We made significant improvements to the paper and re-structured it. Key findings were further elaborated in the new discussions section. Key findings and concluding remarks were presented in the newly created conclusion section. Table 1 and 2 were presented with uniform font sizes.
Comment (4): What is the significance of this work?
Response: This work updated and validated a model that is able to predict lettuce crop growth and presents about the use of several decision support tools, co-optimization of environmental variables evaluated by various what-if scenarios, considering air temperature, carbon dioxide, and daily light integral on crop growth and yield, that can lead to resource and cost savings for lettuce crop production, with also potential application opportunities for other crops, in vertical farming systems. These were articulated clearly in the related sections in the revised paper.
Comment: (5) Mention useful new knowledge about a completely novel idea or it is modifications in existing methods by applying a new algorithm.
Response: Indeed, we discuss in detail the process and modification of a model that can predict lettuce crop yield and demonstrate its strategic use for co-optimization of environmental variables, leading to potential resource and energy savings, which is unique in this paper and research conducted in this study.
Comment (6): How you work is useful to provide directions for future research.
Response: The current work demonstrated improvement of an existing model for its use on predicting crop yield and growth and its strategic use to identify co-optimization of environmental variables, leading to potential resource and energy savings. Future research can consider the methodologies presented in the current paper and evaluate other alternative strategies extending the possibilities of co-optimization of environmental variables, with other crops and crop varieties, that are applicable for vertical farming.
Comment (7): Authors are suggested to review more new and relevant research to support their research contribution. The following references are recommended for possible consideration: Gadekallu, T. R., Srivastava, G., Liyanage, M., Iyapparaja, M., Chowdhary, C. L., Koppu, S., & Maddikunta, P. K. R. (2022). Hand gesture recognition based on a Harris hawks optimized convolution neural network. Computers and Electrical Engineering, 100, 107836
Response: Several additional references were reviewed and cited in the revised paper especially those related to vertical farming systems and environmental controls which is the main focus in the paper.
The following are addition revisions made in the revised paper:
- line 39, added indent
- line 210, table reformatted, reduced font size from 12 to 10, changed font to palatino linotype, adjusted column sizes, removed asterisk, shortened post-text, added vinculums to indicate precision
- line 240, reduced font size from 12 to 10, changed font to palatino linotype
- line 244, removed carriage return
- line 316, table reformatted, reduced font size form 12 to 10, changed font to palatino linotype, adjusted column sizes, removed asterisk, shortened post-text, added vinculums to indicate precision
- line 747, changed section number from 1 to 5
- line 857, added [12] to citation, removed parenthetical citation
- replaced all figures with higher resolution versions
- changed font to palatino linotype for all figure text
- removed whitespace and changed images to anchor to paragraph
- moved text from after figure 4 to between figures 3 and 4, added paragraph commenting on accuracy after figure 4
Respectively submitted,
Murat Kacira
Corresponding author
Reviewer 3 Report
The manuscript is interesting and well written. Several issues regarding format should be improved. For example tables are not well formated, several figures are unclear and the fonts are very small.
The literature review could be expanded considering 10.3390/app13010014 and 10.1016/j.compag.2022.106993.
Please improved conclusions including guidelines for practice as well.
Author Response
Responses to Reviewer Comments
Reviewer 3
Comment: The manuscript is interesting and well written. Several issues regarding format should be improved. For example tables are not well formated, several figures are unclear and the fonts are very small.
Response: We thank the reviewer for the kind comment and constructive review and suggestions. Tables were revised and format was improved. We significantly improved the figures with larger fonts and content presented.
Comment: The literature review could be expanded considering 10.3390/app13010014 and 10.1016/j.compag.2022.106993.
Response: Several additional references, including the one suggested by the reviewer, were added to the revised paper and cited especially those related to vertical farming systems and environmental controls.
Comment: Please improved conclusions including guidelines for practice as well.
Response: The structuring of paper was significantly improved. A discussion and a new conclusions section were created presenting key outcomes and findings of the study. We discussed and presented several decision support tools and environmental control strategies that can be considered for practice and applications in vertical farming systems, which can support creating a control system capable of taking advantage of fluctuations in the price of resources and improving the energy use efficiency of growing crops in vertical farms.
The following are addition revisions made in the revised paper:
- line 39, added indent
- line 210, table reformatted, reduced font size from 12 to 10, changed font to palatino linotype, adjusted column sizes, removed asterisk, shortened post-text, added vinculums to indicate precision
- line 240, reduced font size from 12 to 10, changed font to palatino linotype
- line 244, removed carriage return
- line 316, table reformatted, reduced font size form 12 to 10, changed font to palatino linotype, adjusted column sizes, removed asterisk, shortened post-text, added vinculums to indicate precision
- line 747, changed section number from 1 to 5
- line 857, added [12] to citation, removed parenthetical citation
- replaced all figures with higher resolution versions
- changed font to palatino linotype for all figure text
- removed whitespace and changed images to anchor to paragraph
- moved text from after figure 4 to between figures 3 and 4, added paragraph commenting on accuracy after figure 4
Respectively submitted,
Murat Kacira
Corresponding author
Round 2
Reviewer 1 Report
Dear Authors,
You improved the manuscript.
The list of references still needs augmentation.
I wish you all the best!
Author Response
Responses to Reviewer Comments
Reviewer 1
Reviewer Comment: Dear Authors, you improved the manuscript. The list of references still needs augmentation. I wish you all the best!
Response: Several relevant literatures were added as suggested with discussions in the revised version.
Respectively submitted,
Murat Kacira
Corresponding author
Reviewer 3 Report
The manuscript is improved.
The introduction does not include a literature review of the topic.
It is proposed to consider some more literature like 10.3390/app13010014, 10.1016/j.compag.2022.106993 etc.
Author Response
Responses to Reviewer Comments
Reviewer 3
Reviewer Comment: The manuscript is improved. The introduction does not include a literature review of the topic. It is proposed to consider some more literature like 10.3390/app13010014, 10.1016/j.compag.2022.106993 etc.
Response: Several relevant literatures were added in the introduction section, including the specific references suggested by the reviewer, with discussions in the revised version.
Respectively submitted,
Murat Kacira
Corresponding author